# Late-time correlators and complex geodesics in de Sitter space

Lars Aalsma[1⋆], Mir Mehedi Faruk[2,3,4†], Jan Pieter van der Schaar[2,3‡],
Manus Visser[5°] and Job de Witte[2,3§]

**1** Department of Physics and Beyond: Center for Fundamental Concepts in Science,
Arizona State University, Tempe, Arizona 85287, USA
**2** Institute of Physics, University of Amsterdam, Science Park
904, PO Box 94485, 1090 GL Amsterdam, The Netherlands
**3** Delta Institute for Theoretical Physics, Science Park 904,
PO Box 94485, 1090 GL Amsterdam, The Netherlands
**4** Department of Physics, McGill University, Montreal, QC, H3A 2T8, Canada
**5** Department of Applied Mathematics and Theoretical Physics, University of Cambridge,
Wilberforce Road, Cambridge CB3 0WA, UK

⋆ laalsma@asu.edu , † mir.faruk@mail.mcgill.ca ,
‡ j.p.vanderschaar@uva.nl , ° mv551@cam.ac.uk , § job.de.witte@zeelandnet.nl

## Abstract

We study two-point correlation functions of a massive free scalar field in de Sitter space using the heat kernel formalism. Focusing on two operators in conjugate static patches we derive a geodesic approximation to the two-point correlator valid for large mass and at late times. This expression involves a sum over two complex conjugate geodesics that correctly reproduces the large-mass, late-time limit of the exact two-point function in the Bunch-Davies vacuum. The exponential decay of the late-time correlator is associated to the timelike part of the complex geodesics. We emphasize that the late-time exponential decay is in tension with the finite maximal entropy of empty de Sitter space, and we briefly discuss how non-perturbative corrections might resolve this paradox.

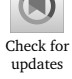

# 1  Introduction

In recent years, we have seen remarkable progress in our understanding of the role that non-perturbative effects play in black hole physics [1–3]. Most prominently, new saddles to the Euclidean gravitational path integral have been found that contribute to the entropy of Hawking radiation [4,5]. When included, these saddles cap off the growth of the entropy of Hawking radiation, thereby showing behavior consistent with unitarity.[1] These results highlight the importance of non-perturbative effects in gravitational backgrounds. In addition, such effects have been investigated in the context of cosmological spacetimes, although there is debate about the existence and nature of the corrections to the entropy of the Gibbons-Hawking radiation in de Sitter space [7–13]. Therefore, the implications of unitary evolution are less clear in cosmology.

Understanding the role that non-perturbative effects play in asymptotically de Sitter space is an especially pressing question, given its direct relevance to our own universe. Although some progress in this direction has been made by studying the behavior of entropy in simplified two-dimensional models of cosmology, ideally one would like to consider more relevant objects, like correlators, in realistic higher-dimensional cosmological spacetimes.

In this paper, we take a modest step in this direction by considering the late-time behavior of certain correlation functions, which is expected to be in conflict with the finite entropy of de Sitter space. In [14] Maldacena argued in the context of AdS/CFT that two-point correlation functions in the thermofield double state of two maximally entangled CFTs show quasi-periodic behavior as a direct consequence of the discrete spectrum of energy eigenstates. Instead, when computing the same correlator in the dual eternal black hole geometry one finds that it exponentially decays. Maldacena suggested that this tension can be resolved by realizing that the duality between the CFT thermofield double state and the bulk geometry prepared by a Euclidean path integral involves a sum over bulk geometries. Typically, the eternal black hole is the dominant saddle point of this path integral, giving rise to exponentially decaying correlation functions, but at late times other saddles, for instance thermal AdS, might become relevant. Although later work [15] showed that thermal AdS cannot completely resolve this tension, it does emphasize the importance of non-perturbative effects, and more recent developments have suggested that (replica) wormholes could resolve this tension.

The motivation for this work is that an analogous tension, between exponentially decaying bulk correlation functions and finite entropy, arises in de Sitter space. The static patch of de Sitter space has a cosmological horizon with an associated entropy proportional to the horizon area in Planck units, $A/4$ [16]. Moreover, it also appears that vacuum de Sitter space corresponds to the maximum entropy state, i.e. any additional energy lowers the entropy. Although the microscopic nature of the de Sitter entropy is still very much mysterious, this does suggest that the Hilbert space of quantum gravity in de Sitter space, and consequently

---

[1]See [6] for an accessible review of these developments.

its holographically dual description, is finite-dimensional [17–21]. Of course, the absence of a well-understood holographic dual to de Sitter space obstructs a direct comparison between correlation functions in the bulk and its dual. Still, assuming the underlying microscopic theory has a discrete energy spectrum, correlation functions in de Sitter space are expected to display quasi-periodic behavior. In lowest order, however, we find that correlation functions with operators in conjugate static patches of de Sitter space decay exponentially at late times.

Specifically, in this paper we consider bulk two-point correlation functions of massive scalar fields, and we compute them by making use of heat-kernel methods. By exploiting an expansion of the heat kernel valid for heavy fields, we derive an expression for the correlator on a generic curved background that involves a sum over geodesics between the two operators. Specifying to de Sitter space and focusing on correlation functions with operators in conjugate static patches, we show that even in the absence of real geodesics connecting these operators, a sum over complex geodesics correctly reproduces the late-time, large mass behavior of the correlator in general dimensions. It was previously observed in [22] that the two-point function in $(2+1)$-dimensional de Sitter space can be expressed as a sum over complex geodesics, but we generalize that result to higher dimensions. We also verify that the sum of the two complex conjugate geodesics indeed shows exponential decay at late times, governed by the real (timelike) part of the complex geodesic. Finally, we discuss the important difference with respect to the AdS eternal black hole and speculate how non-perturbative corrections due to other saddle geometries might modify this result in a manner consistent with finite entropy.

The rest of this article is organized as follows. In Section 2 we set up notation and discuss geodesics in de Sitter space. We use the Schwinger-deWitt formalism to derive a geodesic approximation for the Feynman propagator in a general background. In Section 3 we restrict to propagators in de Sitter space and show how they can be reproduced by employing a geodesic approximation that sums over geodesics. We conclude in Section 4 with a discussion of our results and their relevance for the information paradox in de Sitter space.

**Note added:** We coordinated submission on arXiv with the work [23], which also uses complex geodesics in de Sitter space to compute correlation functions. Our methods are complimentary, since they use a saddle point approximation of a path integral, whereas we employ heat kernel expansion methods.

## 2 Geodesic approximation to correlators in de Sitter space

We start with a review of geodesics in de Sitter space, after which we introduce the Schwinger-deWitt formalism to compute Feynman propagators.

### 2.1 Preliminaries on geodesics in de Sitter space

It is convenient to study geodesics in de Sitter space using the embedding formalism. De Sitter space can be described by the embedding equation (see e.g. [24])

$$\eta_{AB}X^A X^B = \ell^2, \tag{1}$$

where $X^A$ with $A = 0, \ldots, d+1$ are the embedding coordinates, $\eta_{AB}$ is the $(d+2)$-dimensional Minkowski metric with signature $(-+\cdots+)$ and $\ell$ is a length scale called the de Sitter radius. Given two points $(x, y)$ the (square of the) de Sitter invariant distance measured in de Sitter units is given by

$$Z(x, y) = \frac{1}{\ell^2}\eta_{AB}X^A(x)X^B(y). \tag{2}$$

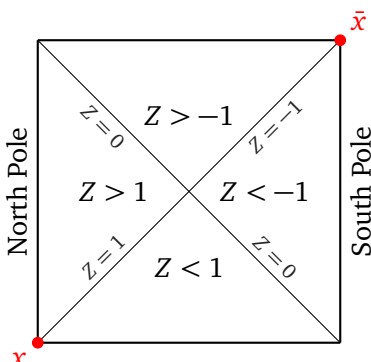

Figure 1: Penrose diagram of de Sitter space with the value of the (dimensionless) de Sitter invariant distance $Z(x, y)$ depicted, where the point $x$ is located in the lower left corner of the Penrose diagram. Further, $\bar{x}$ denotes the antipodal point. For instance, $Z(x, y) > 1$ if the point $y$ lies in the Northern static patch.

Let $\bar{x}$ denote the antipodal point of $x$. The invariant distance $Z(x, y)$ changes sign under an antipodal transformation

$$Z(x, \bar{y}) = -Z(x, y). \tag{3}$$

From the definition (2) it is also clear that $Z(x, y) = Z(y, x)$. The value of the invariant distance depends on the nature of the two points in the $Z \geq 0$ region, i.e.

- $Z(x, y) = 0$ when $x$ is halfway between $y$ and $\bar{y}$,

- $Z(x, y) > 1$ when $x$ and $y$ are timelike separated,

- $Z(x, y) = 1$ when $x$ and $y$ are null separated,

- $Z(x, y) < 1$ when $x$ and $y$ are spacelike separated.

By applying the antipodal transformation on $y$ the value of $Z$ changes to

- $Z(x, y) > -1$ when $x$ and $\bar{y}$ are spacelike separated,

- $Z(x, y) = -1$ when $x$ and $\bar{y}$ are null separated,

- $Z(x, y) < -1$ when $x$ and $\bar{y}$ are timelike separated.

In Figure 1 we denote the value of $Z(x, y)$ in the Penrose diagram of de Sitter space when one of the points lies at the lower left corner. $Z$ is positive in the lower half and negative in the upper half of the Penrose diagram, where we define upper and lower by the horizon at $Z = 0$ that runs from the left upper corner to the right lower corner. The geodesic distance $D(x, y)$ between two points is related to $Z(x, y)$ by

$$\cos\left(D(x, y)/\ell\right) = Z(x, y). \tag{4}$$

This equation shows that the geodesic distance is only real when $|Z(x, y)| \leq 1$ and, in general, complex. We use the convention that $D(x, y)$ can refer to both spacelike and timelike separated points. The former corresponds to real and the latter to imaginary distance. Later on, we will separately discuss spacelike and timelike trajectories, for which we use a different symbol. The fact that $D(x, y)$ can be imaginary has important consequences when we want to connect a pair of points $(x, y)$ by a geodesic. By taking the point $x$ to lie at the intersection of the Northern pole and the (past) cosmological horizon, as indicated in Figure 1, we see that for

any point $y$ in the South Pole static patch there is no real geodesic connecting the two points except at $Z(x,y) = -1$. This corresponds to antipodal points.

From the embedding metric $ds^2 = \eta_{AB} dX^A dX^B$ one can derive the following global de Sitter metric in $d+1$ spacetime dimensions

$$ds^2 = -d\tau^2 + \ell^2 \cosh^2(\tau/\ell) d\Omega_d^2. \tag{5}$$

The topology of Lorentzian de Sitter spacetime is $R \times S^d$. Another useful solution to the embedding equation is de Sitter space in static coordinates, for which the metric reads

$$ds^2 = -f(r) dt^2 + f(r)^{-1} dr^2 + r^2 d\Omega_{d-1}^2, \tag{6}$$

where

$$f(r) = 1 - r^2/\ell^2. \tag{7}$$

Finally, let us consider the explicit expression in Kruskal coordinates, which also give a global cover of de Sitter space. These are related to the static coordinates (in a particular static patch) by

$$x^\pm = \pm \ell e^{\pm t/\ell} \sqrt{\frac{\ell - r}{\ell + r}}. \tag{8}$$

Taking this patch to be centered at the South Pole (right quarter of the Penrose diagram), where time flows upwards, we can define static coordinates in the other quarters of the diagram (Milne patches and Northern static patch) by sending $t \to t + i\epsilon$, where $\epsilon = -\pi\ell$ covers the Northern static patch and $\epsilon = \pm\frac{\pi}{2}\ell$ covers the bottom and top Milne patch, respectively [25]. For $(d+1)$-dimensional de Sitter space these coordinates are related to the embedding coordinates as follows

$$\begin{aligned}
X^0 &= \ell^2 \left( \frac{x^+ + x^-}{\ell^2 - x^+ x^-} \right), \\
X^d &= \ell^2 \left( \frac{x^+ - x^-}{\ell^2 - x^+ x^-} \right), \\
X^i &= \ell \left( \frac{\ell^2 + x^+ x^-}{\ell^2 - x^+ x^-} \right) \omega^i.
\end{aligned} \tag{9}$$

Here, $\omega^i$ with $i = (1, \ldots, d)$ are the coordinates for a unit $(d-1)$-sphere. Let us now assume that two points $x$ and $y$ are each in a conjugate static patch at their respective poles, i.e. $x^+ x^- = y^+ y^- = -\ell^2$. We then find

$$Z(x,y) = \frac{x_+^2 + y_+^2}{2x^+ y^+}. \tag{10}$$

For $x$ at the origin of the Northern static patch and $y$ at the origin of the Southern static patch we have

$$x^+ = -\ell e^{t_x/\ell}, \qquad y^+ = \ell e^{t_y/\ell}, \tag{11}$$

such that

$$Z(x,y) = -\cosh\left( \frac{t_x - t_y}{\ell} \right). \tag{12}$$

Thus, for the endpoints located at the origin of conjugate static patches, which we refer to as podal points, we find that $Z(x,y) \leq -1$ and the only real geodesic connecting the two points satisfies $t_x = t_y$, for which the geodesic distance takes the value

$$D(t_x - t_y = 0) = \pi\ell. \tag{13}$$

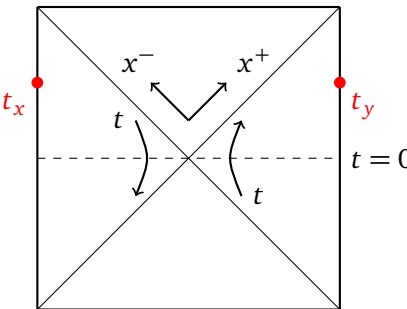

Figure 2: We can use the de Sitter time-translation invariance to map two points on the poles of the spatial sphere to a global equal time slice with $t_x = -t$ and $t_y = t$. There are no real geodesics between these two points unless $t = 0$.

This answer was to be expected since it is just half the circumference of a sphere. The fact that all endpoints with $t_x - t_y = 0$ have the same geodesic distance reflects the invariance of global de Sitter space under a simultaneous time translation in both static patches (upwards in the Southern patch and downwards in the Northern patch).

For more general podal points, we note that without loss of generality we can always define one of the points to lie at $t = 0$. We then find it convenient to use the time-translation invariance to map the points to the symmetric situation where they are located at the same global time slice, see Figure 2. We conclude this section by noting that in general, unless the two points are antipodal, i.e. $t = 0$, no (real) geodesic exists.[2]

## 2.2 Schwinger-deWitt formalism

Now that we have reviewed some essentials of geodesics in de Sitter space, we are ready to compute two-point correlation functions of a massive scalar field using a geodesic approximation scheme. Employing the Schwinger-deWitt proper time formalism, we will obtain an expression valid for massive fields in general dimensions that involves a geodesic between the location of the field operators. A similar derivation can be found in [27, 28].

The starting point is the action for a minimally-coupled real scalar field in $d + 1$ dimensions

$$I = -\frac{1}{2} \int \mathrm{d}^{d+1}x \sqrt{-g} \left( (\partial \phi)^2 + m^2 \phi^2 \right). \tag{14}$$

The equation of motion for the scalar field is the Klein-Gordon equation

$$\left( \Box - m^2 \right) \phi = 0. \tag{15}$$

By supplying the appropriate boundary conditions we can construct the different correlation functions or propagators. The time-ordered Green's function, or Feynman propagator, $G(x, y)$ of the Klein-Gordon operator satisfies

$$\left( \Box - m^2 \right) G(x, y) = -\frac{\delta^{d+1}(x - y)}{\sqrt{-g(x)}}. \tag{16}$$

To write down a useful expression for $G(x, y)$, we view it as the expectation value of an operator $\hat{G}$ in some Hilbert space

$$G(x, y) := \langle x | \hat{G} | y \rangle. \tag{17}$$

---

[2]The same result was found in [26], with the difference that they took the static time to flow upwards in both North and South pole, which flips the sign of one of the time coordinates.

This Hilbert space obeys a completeness relation

$$\int \mathrm{d}^{d+1}x \sqrt{-g(x)}\,|x\rangle\langle x| = \mathbb{1}\,, \tag{18}$$

and orthonormality

$$\langle x|y\rangle = \frac{\delta^{d+1}(x-y)}{\sqrt{-g(x)}}\,. \tag{19}$$

Writing the Klein-Gordon operator as

$$\hat{H} = -\Box + m^2\,, \tag{20}$$

it obeys

$$\hat{H}\hat{G} = \mathbb{1}\,, \tag{21}$$

from which (16) can be derived by taking an expectation value. In the proper time formalism, we can now write a formal expression for this propagator as

$$G(x,y) = i\int_0^\infty \mathrm{d}s\,\langle x|e^{-is\hat{H}}|y\rangle\,. \tag{22}$$

Convergence of this integral requires the imaginary part of $\hat{H}$ to be negative, so we should deform $\hat{H} \to \hat{H} - i\epsilon$ and take $\epsilon \to 0^+$ after evaluating. With this choice the Green's function represents the Feynman propagator [27]. From this expression, one can view $\hat{H}$ as a Hamiltonian that generates time evolution with $s$. Thus, writing

$$|x,s\rangle = e^{is\hat{H}}|x\rangle\,, \tag{23}$$

we see that a general state $|\psi\rangle$ can be expressed in terms of a wave function

$$\psi(x,s) = \langle x,s|\psi\rangle = e^{-isH}\langle x|\psi\rangle\,, \tag{24}$$

which therefore obeys the time-dependent Schrödinger equation

$$i\partial_s\psi(x,s) = H\psi(x,s)\,. \tag{25}$$

We thus reformulated solving (16) as the problem of solving for the motion of a non-relativistic quantum particle on a curved background. In particular, we are interested in computing

$$K(x,y;s) := \langle x|e^{-is\hat{H}}|y\rangle\,, \tag{26}$$

which obeys the Schrödinger equation, subject to the boundary condition

$$\lim_{s\to 0} K(x,y;s) = \frac{\delta^{d+1}(x-y)}{\sqrt{-g(x)}}\,, \tag{27}$$

and yields $G(x,y)$ via (22)

$$G(x,y) = i\int_0^\infty \mathrm{d}s\,K(x,y;s)\,. \tag{28}$$

The quantity $K(x,y;s)$ is known as the *heat kernel* and its evaluation on a generic background is typically complicated. One can use several techniques to evaluate the kernel.

One approach is to write $K(x,y;s)$ as a path integral describing the different paths connecting the endpoints of $(x,y)$ [27]. The path integral can then be evaluated using a saddle point approximation along geodesic paths. This approach was used in [23] to compute correlation functions in de Sitter space. Another approach is to expand for small $s$ in a so-called heat kernel expansion. The coefficients at different orders in this expansion are known as Seeley-deWitt coefficients, which have been computed up to high order and for a variety of fields (see [29] for a detailed review). Exact results for the heat kernel are known for backgrounds with a large degree of symmetry. We follow the second approach and determine the heat kernel to leading order in the $s$-expansion, which will turn out to be sufficient.

## 2.3 Heat kernel expansion

Let us introduce the heat kernel expansion by first studying $K(x, y; s)$ in flat space and subsequently generalizing it to an arbitrary curved spacetime. It is well known that in flat space, the solution to the differential equation

$$i\partial_s K_{\text{flat}}(x, y; s) = H K_{\text{flat}}(x, y; s),$$ (29)

subject to the boundary condition (27) is

$$K_{\text{flat}}(x, y; s) = -i \left(\frac{1}{4\pi i s}\right)^{\frac{d+1}{2}} e^{-im^2 s + \frac{i}{2s}\sigma(x, y)},$$ (30)

as can be checked by direct substitution. Here $\sigma(x, y)$ is defined as one half times the geodesic distance $L(x, y)$ squared, $\sigma(x, y) := \frac{1}{2} L^2(x, y)$. For a geodesic parametrized by $s$ the geodesic distance $L(x, y)$ is given by

$$L(x, y) = \int_{s(x)}^{s(y)} ds' \sqrt{g_{ab} \frac{dx^a}{ds'} \frac{dx^b}{ds'}}.$$ (31)

Instead of working with (31) it is more convenient to define a geodesic path as an extremum of the integral

$$E(x, y) = \frac{1}{2} \int_{s(x)}^{s(y)} ds' \left(g_{ab} \frac{dx^a}{ds'} \frac{dx^b}{ds'}\right).$$ (32)

Evaluated on extremal paths (geodesics) these two integrals are related by $L^2 = 2sE$, so we see that $\sigma(x, y) = sE(x, y)$. From this definition we note that positive $\sigma(x, y)$ corresponds to spacelike separated points and negative $\sigma(x, y)$ to timelike separated points. When we are considering purely spacelike or timelike trajectories we will denote

$$\begin{aligned}
\text{Spacelike:} \quad & \sigma = +\frac{1}{2}\mathcal{D}^2, \\
\text{Timelike:} \quad & \sigma = -\frac{1}{2}\mathcal{T}^2.
\end{aligned}$$ (33)

These quantities are related to $Z(x, y)$ as

$$\begin{aligned}
\text{Spacelike:} \quad & Z = \cos(\mathcal{D}/\ell), \\
\text{Timelike:} \quad & Z = \cosh(\mathcal{T}/\ell).
\end{aligned}$$ (34)

Here $\mathcal{D}$ and $\mathcal{T}$ correspond to the real proper distance and proper time, respectively, for points that are connected by a real geodesic. Later on, when we consider complex geodesics with, for example, complex proper time we will indicate this with a subscript as $\mathcal{T}_c$.

In a general curved background, the heat kernel is modified even at lowest order in the $s$-expansion [29]. In this case, we can write the following ansatz

$$K(x, y; s) = -i \left(\frac{1}{4\pi i s}\right)^{\frac{d+1}{2}} e^{-im^2 s + \frac{i}{2s}\sigma(x, y)} \Delta(x, y)^{1/2} F(x, y; s).$$ (35)

Here, $\Delta(x, y)$ is the $s$-independent Van Vleck-Morette determinant, defined as

$$\Delta(x, y) := \frac{\text{Det}\left(-\frac{\partial^2 \sigma(x, y)}{\partial x^a \partial y^b}\right)}{\sqrt{g(x)g(y)}},$$ (36)

where $g(x)$ is the determinant of the metric at the point $x$. Sometimes the quantity $\Delta$ is called the Van Vleck-Morette biscalar, see e.g. [27], but for simplicity we will refer to it as the Van Vleck-Morette determinant. The interpretation of the Van Vleck-Morette determinant can be understood as follows. We can also represent the heat kernel as a path integral describing the propagation of a massive (quantum-mechanical) point particle on a curved background. To solve this path integral we can use a saddle-point approximation and the one-loop quantum determinant, understood as the path integral over the Gaussian fluctuations around the saddle, is in fact directly related to $\Delta(x, y)$. So the Van Vleck-Morette determinant effectively captures the leading corrections in the quantum mechanical point particle description.

Now $F(x, y; s)$ can be solved for using an iterative procedure, in an expansion in small $s$, by plugging the ansatz for the heat kernel into the Schrödinger equation [29, 30]. This yields a power-law expansion of the form

$$F(x, y; s) = \sum_{n=0}^{\infty} s^n f_{2n}(x, y). \tag{37}$$

Since $s$ parametrizes the geodesic in the quantum-mechanical description, the small $s$ expansion is valid for points $(x, y)$ that are close to each other [27]. Further because the Van Vleck determinant is equal to one in flat space, comparing (35) with the known heat kernel in flat space (30) fixes the leading coefficient $f_0$ in the heat kernel expansion. The fact that this coefficient does not change in curved space is a consequence of the boundary condition (27).[3] We can therefore write [29]

$$K(x, y; s) = -i \left( \frac{1}{4\pi i s} \right)^{\frac{d+1}{2}} e^{-im^2 s + \frac{i}{2s}\sigma(x,y)} \Delta(x, y)^{1/2} (1 + \mathcal{O}(s)). \tag{38}$$

The above expression implicitly assumes that there is only a single geodesic that connects the two points, as is the case in Minkowksi space. However, for a general curved spacetime there can be multiple geodesics. In particular, because the spatial slices of de Sitter space are compact, the assumption of just one geodesic saddle contributing to the path integral is almost certainly incorrect. For instance, if we consider two antipodal points at $t = 0$ there are multiple ways of "going around the sphere" to connect the two points. Nonetheless, for points that are sufficiently close to each other, the path with longer geodesic length is expected to be subdominant since those will yield a larger $s$. As we will see, there is a case of specific interest where multiple geodesics become important.

## 3 Late-time correlators in de Sitter and complex geodesics

We now use the derived heat kernel expansion of the Feynman propagator and specify to de Sitter space to obtain a simple expression for the de Sitter propagator that depends on the geodesic distance. For points that are either timelike separated or connected by a spacelike geodesic we show that gives a good approximation to the exact result for large mass. We then propose a generalization of this result for points that lie in conjugate static patches. Although there are no real geodesics connecting these points, we show that in the late-time limit a sum over complex geodesics exactly reproduces the propagator.

---

[3]By matching onto the flat space result, we are imposing that the heat kernel has the same short-distance singularity as for the flat space vacuum. In de Sitter space, this selects the Bunch-Davies vacuum.

## 3.1 Single geodesic

Using the expression for the heat kernel at leading order in $s$, we perform the integral of (28) to obtain the propagator.[4]

$$G_S(x,y) \simeq N_d \, m^{\frac{d-1}{2}} \sqrt{\Delta(\sigma)} \sigma^{\frac{d-1}{4}} \mathbf{K}_{\frac{d-1}{2}}(m\sqrt{2\sigma}), \tag{39}$$

where

$$N_d = -2^{-\frac{1+3d}{4}} \pi^{-\frac{d+1}{2}} i. \tag{40}$$

Here $\mathbf{K}_a(z)$ denotes the modified Bessel function of the second kind and we explicitly introduced the subscript $S$ on the propagator to indicate that we are considering a single geodesic. We stress that this expression is general and the dependence on the background only appears through the determinant $\Delta(\sigma)$ and the precise expression for the geodesic distance. However, to obtain (39) we neglected higher-order terms in $s$ in the heat kernel, which give extra contributions on the right-hand side proportional to the associated heat kernel coefficients at that order.

Specifying now to de Sitter space, the Van Vleck-Morette determinant takes the form[5]

$$\Delta(\sigma) = \left[ \sqrt{\frac{2\sigma}{\ell^2}} \csc\left( \sqrt{\frac{2\sigma}{\ell^2}} \right) \right]^d. \tag{41}$$

We note that (41) contains singularities at

$$\sigma = \left( \frac{\pi k \ell}{\sqrt{2}} \right)^2 \quad (k \in \mathbb{N}). \tag{42}$$

Clearly, the approximations used to derive (39) must break down at these points and in terms of the de Sitter invariant distance, we find that they correspond to

$$Z(x,y) = \cos(\pi k). \tag{43}$$

The first singularity at $k = 1$ yields $Z(x,y) = -1$, which corresponds to a geodesic distance of $\mathcal{D} = \pi\ell$. This is precisely the location where the two geodesics around the de Sitter sphere give an equal contribution to the propagator such that the single geodesic assumption is no longer valid. We come back to this point later and for now explore the validity of the single-geodesic approximation away from the singular points.

Let us now distinguish points that are either spacelike or timelike separated. As discussed, for large mass equation (39) should give a good approximation to the exact Green's function as long as a single geodesic gives the dominant contribution. We now confirm this by comparing the approximated result with the known exact propagator. To do so, we first consider the Wightman function defined as

$$W(x,y) := \langle \phi(x)\phi(y) \rangle. \tag{44}$$

In the Bunch-Davies vacuum it is given by (see e.g. [32, 33])[6]

$$W(x,y) = \frac{\Gamma(\Delta_+)\Gamma(\Delta_-)}{\ell^{d-1}(4\pi)^{\frac{d+1}{2}}\Gamma\left(\frac{d+1}{2}\right)} {}_2F_1\left( \Delta_+, \Delta_-, \frac{d+1}{2}; \frac{1+Z(x,y)}{2} \right), \tag{45}$$

---

[4]Neglecting higher powers in $s$ is valid for large mass, because the $s$ expansion corresponds to a large mass expansion in the propagator [29].

[5]This can be obtained from analytic continuation on an AdS background [31].

[6]We do not consider more general de Sitter invariant vacua such as $\alpha$-vacua [34, 35] in which the Wightman function contains an additional hypergeometric function that can be obtained by sending $Z \to -Z$.

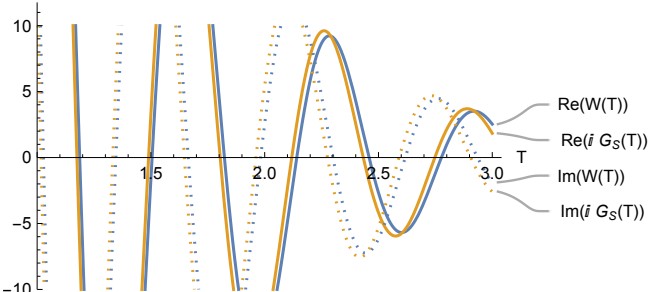

Figure 3: Comparison of the exact (45) and approximated (39) expression for the Green function for points that are timelike separated. We took $m\ell = 10$, $\ell = 1$ and $d = 3$.

where $_2F_1(a, b, c; z)$ is the Gaussian hypergeometric function with

$$\Delta_\pm = \frac{d}{2} \pm i\nu, \quad \text{and} \quad \nu = \sqrt{m^2\ell^2 - \frac{d^2}{4}}. \tag{46}$$

We can introduce the scaling dimension $\Delta$ via (see e.g. [36])

$$m^2\ell^2 = \Delta(d - \Delta), \tag{47}$$

which is solved by $\Delta_\pm$. There are two distinct representations:

1. Complementary series: $0 < \Delta < \frac{d}{2}$  $(0 < m\ell < \frac{d}{2})$,

2. Principal series: $\Delta = \frac{d}{2} + i\nu$  $(m\ell \geq \frac{d}{2})$.

We are mostly interested in heavy scalars and therefore focus on the principal series. The Wightman function is analytic except along the branch cuts at $|Z| \geq 1$. We need to specify how to approach the branch cut. The Feynman propagator is given by taking $Z \to Z + i\epsilon$ with $\epsilon > 0$ such that we have the relation

$$iG(x, y) = W(x, y), \tag{48}$$

with the prescription $Z \to Z + i\epsilon$ understood in the Wightman function [37].

We can compare this exact result with the approximation (39). In Figure 3 and 4 we compare the two expressions for both timelike and spacelike separation respectively, finding excellent agreement away from the singularity at $\mathcal{D} = \pi\ell$.

Next, we also consider the behavior of correlators that receive contributions from multiple geodesics. For this purpose, it is convenient to consider the limit of large proper time: $\mathcal{T}/\ell \gg 1$. By expanding $G_S(x, y)$ for large proper time we obtain

$$\text{Timelike:} \quad G_S(x, y) \simeq -i\sqrt{\frac{m^{d-2}}{4\pi^d\ell^d}} e^{-i\pi\frac{d}{4} - \frac{d}{2}\frac{\mathcal{T}}{\ell}} e^{-im\mathcal{T}} \quad (\mathcal{T}/\ell \gg 1). \tag{49}$$

We also plot the difference between the approximated and exact Green's function in Appendix B.

## 3.2 Multiple geodesics

Let us now discuss the case where multiple geodesics can give a contribution to the propagator. It is natural to expect that the approximation in (39), which only includes a single geodesic,

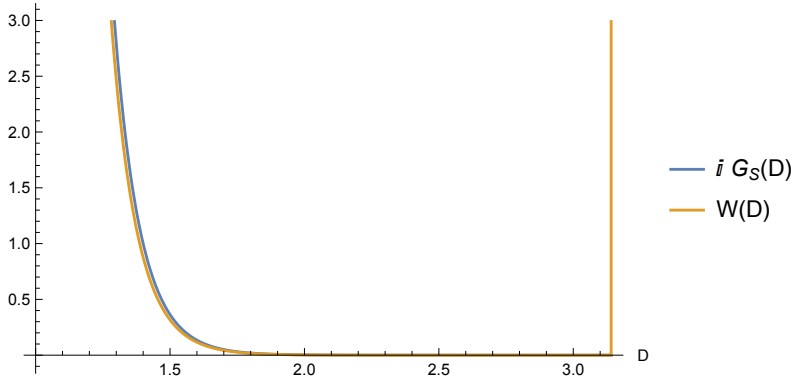

Figure 4: Comparison of the exact (45) and approximated (39) expression for the Green function for points that are spacelike separated. We took $m\ell = 10$, $\ell = 1$ and $d = 3$. The approximated Green function has a singularity at $\mathcal{D} = \pi\ell$, where the single geodesic approximation breaks down.

should be generalized in that case by taking a sum over geodesics. This is relatively easy to see when we want to connect two antipodal points, for example located at $t = 0$ in static coordinates. In this case, there are two real geodesics of equal length corresponding to going clockwise or counter clockwise around the sphere.[7]

When there are multiple geodesics connecting two points, it makes sense to consider additional solutions to the Schrödinger equation governing the heat kernel. Because the Schrödinger equation is linear, a linear combination of solutions is also a solution. As we will see, it is necessary to include a sum over solutions to reproduce the propagator. This picture aligns well with a saddle-point approximation in a path integral representation of the heat kernel. If there are multiple geodesics with a large separation of lengths one will give the dominant contribution. Instead, when multiple geodesics have similar lengths they should be summed over.

It is however less clear how we should interpret the geodesic approximation when we consider the propagator between points in conjugate static patches away from $t = 0$. As discussed, in that case there are no real geodesics connecting these two points. Still, we will show that it is possible to reproduce the heavy-mass and late-time limit of the propagator by summing over complex geodesics that are each others complex conjugate.

To do so, we will employ an expansion of the exact propagator in the limit of large de Sitter invariant distance. Using the asymptotic expression for the hypergeometric function given in Appendix A, we find that the Wightman function naturally splits into two parts in the limit $|z| \to \infty$, where we defined $z := (1 + Z)/2$. Due to the relation $Z = \cos(D/\ell)$, we note that this limit can only be satisfied when $D$ has a non-zero imaginary piece. In that case, we can write

$$\lim_{|z| \to \infty} W(x, y) = \mathcal{W}(\Delta_+, \Delta_-; z(x, y)) + \mathcal{W}(\Delta_-, \Delta_+; z(x, y)), \tag{50}$$

where

$$\mathcal{W}(\Delta_+, \Delta_-; z(x, y)) = \frac{\Gamma(\Delta_+)\Gamma(\Delta_- - \Delta_+)}{(4\pi)^{\frac{d+1}{2}}\ell^{d+1}\Gamma\left(\frac{d+1}{2} - \Delta_+\right)}(-z)^{\Delta_+}. \tag{51}$$

This asymptotic form has an interesting interpretation in the dS/CFT correspondence. This is most transparent in planar coordinates, in terms of which the de Sitter metric takes the form

$$ds^2 = \frac{\ell^2}{\eta^2}\left(-d\eta^2 + d\vec{x}_d^2\right). \tag{52}$$

---

[7]Strictly speaking there are more geodesics, but those can be related by an isometry.

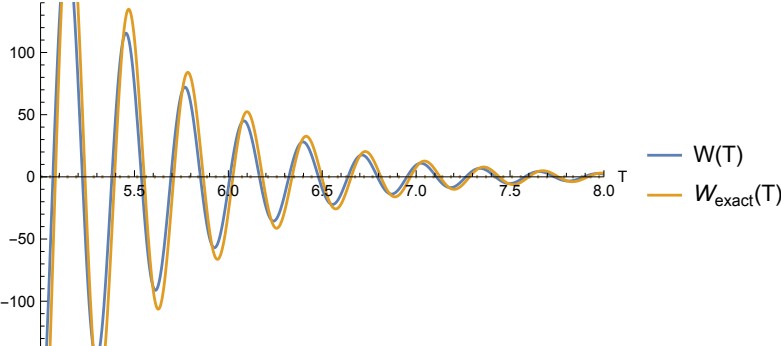

Figure 5: Comparison of the exact (45) and approximated (57) expression for the Wightman function. We took $m\ell = 20$, $\ell = 1$ and $d = 3$.

For operators at the origin of their respective static patches we have $z = \eta/\ell$ at late times. We then see that the Wightman function schematically splits up as follows.

$$\lim_{\eta \to 0} W(x, y) = (\dots)\left(-\frac{\eta}{\ell}\right)^{\Delta_+} + (\dots)\left(-\frac{\eta}{\ell}\right)^{\Delta_-} . \tag{53}$$

Interpreted in terms of a CFT living at $\mathcal{I}^+$, this two-point function receives a contribution from two CFT operators with conformal dimensions $\Delta_\pm$ [38].

Next, we note that for scalars in the principal series representation $(\Delta_+)^* = \Delta_-$, where the star denotes complex conjugation. This implies that when $\arg(z) = \pi$ the two terms are each others complex conjugate. Using the relation between $z$ and $Z$ we notice from Figure 1 that this occurs when two points are located in conjugate static patches in such a way that $(x, \bar{y})$ are timelike separated. Thus, for these correlators we can write

$$W(x, y) = \mathcal{W}(\Delta_+, \Delta_-; z(x, y)) + (\mathcal{W}(\Delta_+, \Delta_-; z(x, y)))^* \quad (\nu \in \mathbb{R}, \arg(z) = \pi) . \tag{54}$$

As we will see, these two terms should be interpreted as two complex geodesics that are each others complex conjugate. To compare this with our expression for the single geodesic, given by (49), we expand for $m\ell \gg 1$. It will then be useful to express $z(x, y)$ in terms of proper time. To do so, let us first consider two timelike separated points in the same static patch. At late times, this corresponds to $z = +e^{\mathcal{T}/\ell}$. From the antipodal map $Z(x, y) = -Z(x, \bar{y})$, we then see that for two points in conjugate static patches this leads to $z = -e^{\mathcal{T}/\ell}$, which has $\arg(z) = \pi$. This effectively corresponds to evaluating at complex proper time $\mathcal{T}_c = -i\pi\ell + \mathcal{T}$. We therefore find

$$\mathcal{W}(\Delta_+, \Delta_-; \mathcal{T}_c) = \sqrt{\frac{m^{d-2}}{4\pi^d \ell^d}} e^{i\pi\frac{d}{4} - m\ell\pi} e^{-\frac{d}{2}\frac{\mathcal{T}}{\ell}} e^{-im\mathcal{T}} . \tag{55}$$

The similarity of $\mathcal{W}(\Delta_+, \Delta_-; \mathcal{T}_c)$ to the expression for a single geodesic given by (49) is striking. The two expressions differ only by a constant and a factor of $\exp(-m\ell\pi)$. Constructing the total Wightman function by adding the complex conjugate we find

$$W(x, y) = \mathcal{W}(\Delta_+, \Delta_-; \mathcal{T}_c) + (\mathcal{W}(\Delta_+, \Delta_-; \mathcal{T}_c))^* = \sqrt{\frac{m^{d-2}}{\pi^d \ell^d}} e^{-m\ell\pi} e^{-\frac{d}{2}\frac{\mathcal{T}}{\ell}} \cos\left(\frac{d\pi}{4} - m\mathcal{T}\right) . \tag{56}$$

Similar expressions for the two-point function have been considered in [22,39,40], but instead by considering a large-mass expansion. Remarkably, using the relation $W(x, y) = iG(x, y)$ between the Wightman function and the propagator, we see that we can reproduce the late-time and heavy-mass limit of the Wightman function by taking the expression for a single complex

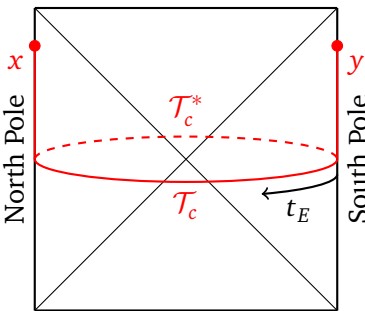

Figure 6: The two complex conjugate paths that connect the points at $x$ and $y$. The Euclidean time $t_E = \text{Im}(\mathcal{T}_c)$ has periodicity $t_E \sim t_E + 2\pi\ell$. The solid arc corresponds to $\text{Im}(\mathcal{T}_c) = +\pi\ell$ and the dashed arc to $\text{Im}(\mathcal{T}_c^*) = -\pi\ell$. Moving along the arc corresponds to evolution along the Euclidean time by either $t_E = \pm\pi\ell$.

geodesic evaluated at proper time $\mathcal{T}_c = -i\pi\ell + \mathcal{T}$ and adding to it its complex conjugate. Concretely, we find

$$W(x, y) = iG_S(\mathcal{T}_c) - i\left(G_S(\mathcal{T}_c)\right)^* . \tag{57}$$

The expression for a single geodesic was given in (49). Importantly, this results holds for all dimensions and for two scalar fields located at arbitrary points in conjugate causal patches, so when $z \ll -1$.[8] We also observe that, previously, in the single geodesic approximation the Van Vleck-Morette determinant contained singularities, signaling a breakdown of the single-geodesic approximation. The divergences can be removed by adding the second geodesic, hence the expression (57) does not contain singularities. On top of this analytical result, we numerically show the agreement between the approximated and exact Wightman function at late times in Figure 5. The difference between the exact and approximated Wightman function is given in Appendix B.

We would like to stress that, although knowing the exact answer for the Wightman function allowed us to easily identify the geodesics that need to be summed over, this is strictly speaking not required. As discussed before, another way of solving the heat equation is to write the heat kernel in terms of a path integral. Solving this path integral using a saddle point approximation then naturally specifies the geodesics that need to be summed over. In fact, this complementary approach was taken in [23]. The expression (57) can therefore be generalized to situations where the exact Wightman function is unknown.

Finally, let us comment on the interpretation of this relation in terms of complex geodesics. For ease of presentation, we find it useful to consider points at the poles of the conjugate static patches (podal points), although we stress that (49) holds more generally. In static coordinates, this corresponds to $r = 0$. As before, without loss of generality we can use the de Sitter isometries to move the operators to a global equal time slice, corresponding to $-\frac{1}{2}t$ at the North pole and $+\frac{1}{2}t$ at the South pole, see Figure 6. Remember that the time coordinate flows in opposite directions at the two poles. For these points, the de Sitter invariant distance is given by $Z = -\cosh(t/\ell)$, which for large $t/\ell$ translates to a complex geodesic invariant length (proper time) of

$$\mathcal{T}_c^* = -i\pi\ell + t . \tag{58}$$

Evolving in both real (Lorentzian) time $t$ and imaginary (Euclidean) time $t_E$, this can be interpreted as a trajectory where we start out at the South pole, flow downwards to $t = 0$, move

---

[8]Note that any two points taken to the future spacelike infinity boundary of de Sitter space will be causally disconnected and should therefore fall into this category, as clearly reflected by the identified behavior of the Wightman function in the limit of vanishing conformal time, cf. (53).

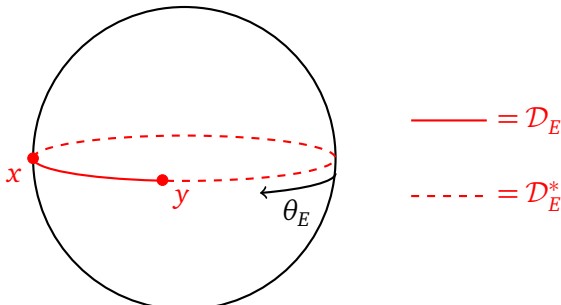

Figure 7: Two points on the Euclidean de Sitter sphere, which are connected by two geodesics indicated by the solid and dashed red lines with length $\mathcal{D}_E$ and $\mathcal{D}_E^*$, respectively. The direction along the red circle is the angle $\theta_E$. We choose the point $x$ to be located at $\pi$ and the point $y$ at some angle $\theta_E$.

halfway along the Euclidean circle (with $t_E = -\pi\ell$) to the North pole and finally evolve downwards again in Lorentzian time. The full correlator now involves the sum of this geodesic and its complex conjugate, see Figure 6.

### 3.3 Euclidean interpretation

The above result can also be derived from a fully Euclidean perspective. Euclidean de Sitter space can be obtained from an analytic continuation of Lorentzian global de Sitter space by sending $\tau \to i\tau_E$, where the Euclidean time is taken to be periodic $\tau_E \sim \tau_E + 2\pi\ell$. The Euclidean de Sitter metric is hence

$$\mathrm{d}s^2 = \mathrm{d}\tau_E^2 + \ell^2\cos^2(\tau_E/\ell)\mathrm{d}\Omega_d^2, \tag{59}$$

which turns into the standard metric on a round sphere $S^{d+1}$ by sending $\tau_E/\ell \to \theta_E - \pi/2$. The same geometry follows from performing an analytic continuation of the static metric. The advantage of working with Euclidean de Sitter space is that there now exist real geodesics between any two points on the sphere. We can then use the geodesic approximation for real geodesics and sum over the different contributions.

For example, consider two points on the sphere that are separated by some angle $\theta_E$, as indicated in Figure 7. The solid and dashed geodesics have geodesic distance, respectively,

$$\mathcal{D}_E = (\pi - \theta_E)\ell, \qquad \mathcal{D}_E^* = (\pi + \theta_E)\ell. \tag{60}$$

If we now continue this back to Lorentzian signature by analytically continuing the distance $\mathcal{T}_c = i\mathcal{D}_E$ and identifying $\theta_E = it$ as evolution in real Lorentzian time we obtain two complex geodesics with proper time

$$\mathcal{T}_c = i\pi\ell + t, \quad \text{and} \quad \mathcal{T}_c^* = -i\pi\ell + t, \tag{61}$$

corresponding to the two conjugate geodesics that we summed over in the Lorentzian analysis. When one geodesic is much longer than the other in the Euclidean geometry it is suppressed, but both geodesics need to be taken into account in Lorentzian signature since they have the same real part. This shows that the same result can be derived by starting from Euclidean de Sitter and defining the two-point correlator there [23]. After an appropriate continuation to Lorentzian signature this then produces the correct expression for the correlator.

# 4 Discussion: Correlators and the information paradox in de Sitter space

The main result of this work was to show that a two-point correlation function of massive scalar fields in de Sitter space can be computed using a geodesic approximation. This generalizes similar approximation schemes studied in AdS/CFT [41–43] to the cosmological, de Sitter, context. We demonstrated that the geodesic approximation in de Sitter space, for large masses and late times, involves complex geodesics, generalizing earlier work [22] to higher dimensions.[9] Looking at the final result for the correlator (56) we note that it is exponentially decaying in time, corresponding to the timelike part of the complex geodesic.

Having established this result, let us briefly discuss its potential relevance in the context of a version of the information paradox in de Sitter space, which was the main motivation for this work. But first let us comment on the subtle role of observables in de Sitter. In contrast to anti-de Sitter space, where the boundary CFT correlation functions are well-defined objects in quantum gravity, the absence of asymptotic regions (where gravity decouples) in de Sitter space implies that a correlation function is not a gauge-invariant observable, since it transforms non-trivially under diffeomorphisms. That means that, strictly speaking, correlation functions in de Sitter space are only sensible quantities in the limit $\ell_p/\ell \ll 1$. Assuming we are working in this limit, we then only need to impose invariance under the isometries preserved by the background, which act as gauge constraints in gravity. As recently pointed out by [45] however, to define a sensible algebra of observables in a de Sitter static patch, one needs to take into account the role of an observer minimally equipped with a clock, effectively breaking the time-translation invariance of the de Sitter background. In this regard we believe the correlators considered in this work can be viewed as sensible physical observables. Indeed, assuming a limit of weak gravity, these correlators evaluate heavy field operators at late (global) times inserted at (the center of) conjugate static patches, which break the time translation symmetry because we evolve both operators forward, i.e. towards future infinity.

Assuming the finite entropy of de Sitter space implies that the microscopic quantum gravity description involves a discrete spetrum of energy eigenstates, that are presumably most effectively captured by a holographic theory,[10] one expects on general grounds that the late-time behavior of correlation functions should be quasi-periodic. This is analogous to Maldacena's original observation in the context of the AdS eternal black hole [14], and also closely related to the inconsistency between finite de Sitter entropy and the absence of a finite-dimensional representation of the de Sitter isometry group [19]. Using a geodesic approximation we have seen that the two-point function (56), besides an oscillating part, is exponentially suppressed at late times, due to the timelike part of the complex geodesic. This is strikingly different from the situation in the AdS eternal black hole, where at least for dimensions smaller than 4 the decay can be traced back to the exponential growth of the spacelike geodesic probing the interior of the black hole [43, 48]. For de Sitter space the exponential decay is instead generated by the timelike contribution, which is not probing the region beyond the de Sitter horizon. This difference might be important to keep in mind when considering non-perturbative gravitational corrections that can produce late-time quasi-periodic behavior.

In the case of the AdS eternal black hole in two dimensions, it was shown in [49] that the contribution of Euclidean wormholes in JT gravity provides the required shortcuts, circumventing (or opening up) the black hole interior, reproducing the expected ramping and plateau behavior of the correlator at late times. More generally, holographic considerations seem to

---

[9]Complex (null) geodesics also play a role in AdS/CFT where they capture the quasi-normal mode spectrum of black holes [44].

[10]Recently, some properties of such a holographically dual description have been investigated by considering a de Sitter version of the Ryu-Takayanagi formula [46, 47].

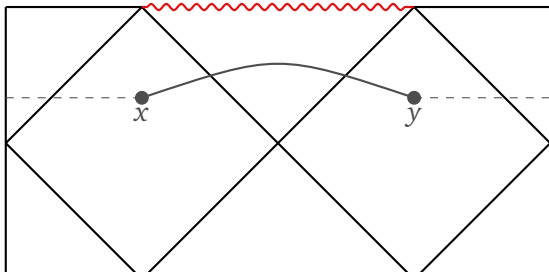

Figure 8: Penrose diagram of Schwarzschild-de Sitter spacetime. The left and right vertical lines are identified. Two points located in conjugate static patches can be connected by a real geodesic through the black hole horizon (solid line) or by a complex geodesic across the cosmological horizon (dashed line).

suggest that non-perturbative gravitational corrections can be sufficient to alter the correlator at late times to be in agreement with finite entropy characteristics. In the case of de Sitter, one example of a non-perturbative (topology changing and de Sitter symmetry breaking [50]) correction to the correlator is the (one-parameter family of) Euclidean Schwarzschild-de Sitter black hole geometries, whose effects can be included using constrained instanton methods [51, 52]. It would clearly be of interest to study the corrections due to these Euclidean de Sitter black holes in terms of the geodesic approximation, which is expected to introduce contributions from additional (complex) geodesics connecting the two points in the different causal patches, see Figure 8 for an example. Naively, it is hard to imagine how these corrections could affect the exponential decay of the correlator, as they do not appear to behave as wormhole (or baby universe) 'shortcuts' in the de Sitter geometry. However, one might speculate that once the effect of de Sitter black hole instantons is understood, more standard 'shortcut' Euclidean wormholes, on top of the Euclidean de Sitter black holes, corresponding to higher order topological transitions, can perhaps be studied (in less than four dimensions) and might halt the exponential decay. We hope this work is a modest first step to start putting these ideas on a firmer footing.

## Acknowledgements

We would like to thank Damián Galante, Edward Morvan and Dong-Gang Wang for helpful discussions.

**Funding information** LA is supported by the Heising-Simons Foundation under the "Observational Signatures of Quantum Gravity" collaboration grant 2021-2818. MRV is supported by SNF Postdoc Mobility grant P500PT-206877 "Semi-classical thermodynamics of black holes and the information paradox". This work is also part of the Delta ITP consortium, a program of the Netherlands Organisation for Scientific Research (NWO) that is funded by the Dutch Ministry of Education, Culture and Science (OCW).

## A  Expansion of hypergeometric function

Here we consider a useful expansion for the Gaussian hypergeometric function. First we use an identify that transforms the argument from $z \to 1/z$, see e.g. [53]

$$
\begin{aligned}
_2F_1(a,b,c;z) = & \frac{\Gamma(c)\Gamma(b-a)}{\Gamma(b)\Gamma(c-a)}(-z)^{-a}\, _2F_1\left(a,a-c+1,a-b+1,\frac{1}{z}\right) \\
& + \frac{\Gamma(c)\Gamma(a-b)}{\Gamma(a)\Gamma(c-b)}(-z)^{-b}\, _2F_1\left(b,b-c+1,b-a+1,\frac{1}{z}\right).
\end{aligned}
\tag{A.1}
$$

We now note the following limit

$$
\lim_{|z|\to\infty} {}_2F_1(a,b,c;1/z) = 1 ,
\tag{A.2}
$$

such that for $|z| \to \infty$ we obtain the expansion

$$
_2F_1(a,b,c,z) \simeq \frac{\Gamma(c)\Gamma(b-a)}{\Gamma(b)\Gamma(c-a)}(-z)^{-a} + \frac{\Gamma(c)\Gamma(a-b)}{\Gamma(a)\Gamma(c-b)}(-z)^{-b} .
\tag{A.3}
$$

## B  Validity of approximated Green's function

In addition to our analytical derivations in the main body, here we also plot the difference between the approximated and exact Green's function for a single geodesic. For spacelike separated points we show that the approximation becomes better for larger mass, see Figure 9. For timelike separated points, in addition, the approximation becomes better for large timelike separation as we demonstrate in Figure 10. Finally, we also consider the difference between the exact and approximated Wightman function for points in conjugate static patches. This approximation becomes better for larger (Lorentzian) timelike separation, see Figure 11.

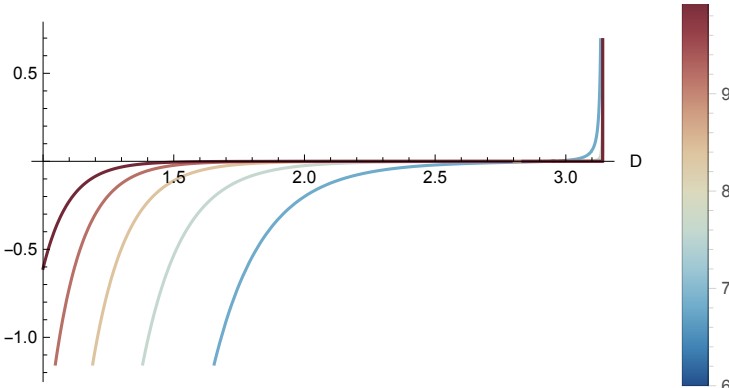

Figure 9: The difference between the exact (45) and approximated (39) expression for the Green's function for spacelike separated points as a function of the proper distance $D$. The color coding indicates the different values of $m\ell$. The approximation becomes better for heavier mass. In the single-geodesic approximation there is a singularity for antipodal points at $D = \pi\ell$.

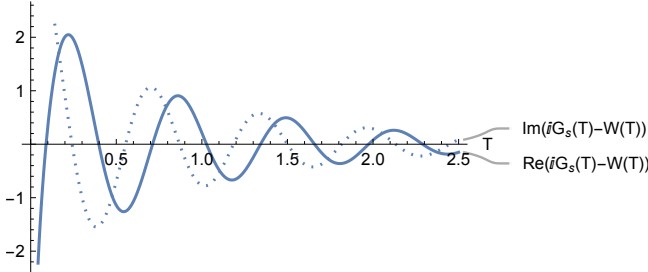

**Figure 10:** The difference between the exact (45) and approximated (39) expression for the Green's function for timelike separated points as a function of the proper time $\mathcal{T}$. The approximation becomes better for larger timelike separation.

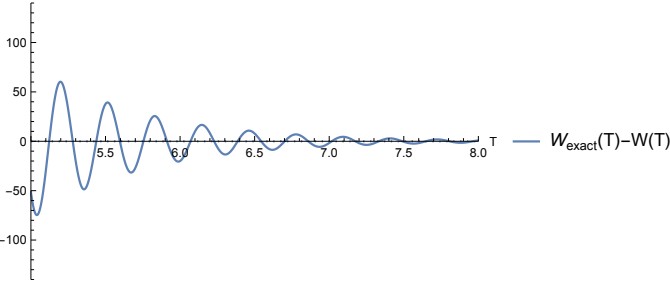

**Figure 11:** The difference between the exact (45) and approximated (57) expression for the Wightman function as a function of proper time $\mathcal{T}$. We took $m\ell = 20$, $\ell = 1$ and $d = 3$. The difference asymptotes to zero for large $\mathcal{T}/\ell$.

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
