# Peer review of "Late-Time Correlators and Complex Geodesics in de Sitter Space"

_SciPost Physics, doi:SciPost Phys. 15, 031 (2023)_

## Round 1 · Referee Report · Anonymous (Referee 1) · 2023-2-3

Strengths

1 - it is clearly presented, with ample background material.

Weaknesses

1 - there is only one novel computation, and it is very similar to what has appeared in a special case before.

Report

This paper studies the geodesic approximation to the two-point correlation function of a massive (free) scalar in de Sitter space. The exact correlator is known, so the main purpose is to show how, in an expansion of the heat kernel, complex geodesics are required to give the right answer. While the paper is mostly review, the point made is an important one so I recommend it be published.

The framing of the paper does not give due credit to the authors' reference [36]. That paper showed the contribution of complex geodesics in the case of dS$_3$, and the purpose of this paper is to generalize that result to higher dimensions. While even before [36] the result of complex geodesics dominating the answer was known, I don't know of a paper where it appears explicitly, so [36] should be cited prominently as being the first appearance of this fact.

Requested changes

1) Modify the referencing of [36]. 2) The last sentence in the first paragraph is inaccurate, in particular the line indicating that these effects are a universal property of event horizons [7-13]". This makes it seem like there are such corrections to the entropy of Hawking radiation of the dS horizon. In the papers [7-13], one of the following things has been done: a dramatic change of the geometry or quantum state ([12-13]), an error in the calculation of fine-grained entropy ([10]), an added rule of subtracting points to obtain nontrivial islands ([9]), islands which are relevant only to the BH region of dS ([8]), and an ambiguity in choice of contour ([7]). I have no idea what [11] is doing. But in any case, the situation for dS is very much unlike the case for AdS black holes, so this line should be modified. 3) (2.31) needs elaboration. The words before it say $\sigma(x,y)$ is half the square of the geodesic distance, but (2.31) seems to be displaying something else (the integrand is squared instead of the total length and there is a $1/s$). 4) The words describing the expansion in (2.36) are a bit confusing. In particular,we now know that the leading term in the $s$-expansion should reduce to the flat space result...where the dots describe higher-curvature terms that are suppressed by powers of $s$." This makes it seem as if the leading piece is the flat space answer and then \emph{all} of the curvature dependence is encoded in the $\dots$. But $\sigma(x,y)$ and $\Delta(x,y)$ both depend on the curvatures, so this is not exactly an expansion in curvatures. 5) On pg. 10 it is stated Thus, for small geodesic distance $\sigma$ they should be subdominant." Isn't $\sigma$ the square of the geodesic distance? And the convention here is that the square of the geodesic distance can be negative, so doessmall" mean small in magnitude? 6) On pg. 11 it says As discussed, for large mass equation (3.1) should give a good approximation to the exact Green function as long as a single geodesic gives the dominant contribution." As far as I could see this isn't discussed/explained earlier in the text. 7) There is a choice made in (3.7) which neglects the second solution for the Wightman function which is a similar hypergeometric function with $Z \rightarrow -Z$ and has the same short-distance structure. (See e.g. https://arxiv.org/pdf/hep-th/0110007.pdf). Eliminating all non-coincident singularities can eliminate this solution but it should at least be mentioned. 8) At the beginning of section 3.2 it saysIt is natural to expect that the approximation (3.11) should involve a sum over geodesics in that case. This is relatively easy to see when we want to connect two antipodal points, for example located at $t=0$ in static coordinates." Doesn't this example have zero proper time separation and therefore break the large-proper-time assumption behind (3.11)? 9) Figure 7 is confusingly drawn. It seems that $t_E$ is the separation between $x$ and $y$ such that $t_E = 0$ puts them on top of each other. But according to (3.22) $t_E = 0$ means they are antipodal points. Further labeling the drawing would help. 10) It's not totally clear what problem is being addressed in the last paragraph of the paper. Are the authors working on a Schwarzschild-dS background, and then arguing that wormholes in the black hole context will allow for shortcuts for the geodesic, as in https://arxiv.org/pdf/1910.10311.pdf? What are the ``(higher-order) corrections" referred to in this paragraph -- higher order in what? Furthermore, the authors assume that there exist real geodesics that span the black hole interior, but at late times such geodesics do not exist.

  • validity: top
  • significance: ok
  • originality: ok
  • clarity: high
  • formatting: excellent
  • grammar: perfect

Author:  Lars Aalsma  on 2023-03-30  [id 3528]

(in reply to Report 1 on 2023-02-03)

We thank the referee for their careful reading of the manuscript and constructive feedback. We have made the following changes that we believe address the referee's comments. A detailed list of changes will be attached to the resubmission.

  1. We modified the referencing to the paper of Fischetti, Marolf and Wall to make it clear that this was the first paper to point out the role of complex geodesics in correlation functions in de Sitter.

  2. We modified this sentence to stress that the status of islands in de Sitter space is quite different than for AdS black holes.

  3. We added additional explanation around (2.31) to clarify the relation between the integral presented in (2.32), which we used to define the geodesic distance, and the standard expression for the geodesic distance given in (2.31).

  4. We improved the explanation of the small s-expansion. We also added footnote 4 to make it clear that higher powers of s in the heat kernel lead to terms that are suppressed by inverse powers of mass in the propagator.

  5. In light of the improved explanation of the small s-expansion we removed this sentence.

  6. We added an explanation of this fact in footnote 4.

  7. We added footnote 6 to make clear that we are focusing on the Bunch-Davies vacuum and do not consider alpha-vacua.

  8. We agree with the referee. In the previous version this statement referred to the wrong equation. We changed this sentence to refer to (3.1), which does not correspond to a large proper time expansion.

  9. We modified Fig. 6 and 7 and distinguished between the Euclidean time coordinate t_E used in Fig. 6 and the Euclidean angle in Fig. 7. The relationship between these two (a priori different) quantities is given around (3.22).

  10. We significantly changed the discussion section. We clarified the difference between the results for the AdS eternal black hole and de Sitter, adding some references to that effect. In addition we clarified our (admittedly) speculative remarks regarding potential non-perturbative corrections that could affect the late-time correlator in de Sitter, including the Euclidean de Sitter black holes instantons. To that effect we also added an additional reference and emphasize that most of the current intuition is based on results in two (or at most three) dimensions.

---

## Round 1 · Referee Report · Anonymous (Referee 2) · 2023-2-21

Report

The authors approximate two-point functions of massive scalars in de Sitter using geodesics.

While the existence of this approximation is not particularly surprising, the authors apply an interesting method, and highlight the role played by complex geodesics. I think the paper should be published for these reasons, meeting the criteria of SciPost Physics.

However there are certain aspects of the work that should be addressed as below.

Requested changes

  1. It is not clear to what extent the authors method is a self-contained derivation of the propagator. i.e. it appears from the discussion surrounding (3.19) that the known answer is used as an input to the calculation, informing which geodesics to include. I would like the authors to clarify if their method can be used to compute the two-point function without using the known answer.

  2. The demonstration of the level of numerical agreement between the approximation and the exact answer needs significant improvement, particularly in light of statements like "we show that a complex geodesic correctly reproduces the propagator in this case.". The main evidence supporting such statements appears to be figures 3,4,5. e.g. for figure 5 the authors state that the agreement is excellent at large T/l, but it looks to me like both curves are getting smaller and it is not obvious that the relative error is decreasing. Similar comments apply to figures 3,4. A more suitable approach would be to indicate the relative error and the convergence towards zero as the approximation is refined.

  • validity: -
  • significance: -
  • originality: -
  • clarity: -
  • formatting: -
  • grammar: -

Author:  Lars Aalsma  on 2023-03-30  [id 3529]

(in reply to Report 2 on 2023-02-21)

We thank the referee for their constructive feedback. We have made the following changes that we believe address the referee's comments. A detailed list of changes will be attached to the resubmission.

  1. We agree with the referee that knowing the exact answer was useful to understand which geodesics to include. Nonetheless, the geodesics that need to be included can be determined by a saddle point approximation. We did not show this explicitly in our manuscript, but have added a reference to the paper [23] where this computation was performed explicitly.

  2. To add more support to our claim that the geodesic approximation gives a good approximation of the exact result we added Appendix B. Here, we give additional numerical evidence that the difference between the approximation and exact result goes to zero in the limit when the approximation becomes better, as suggested by the referee. In addition, we'd like to stress that our main result - complex geodesics reproducing the exact correlator in the limit of large mass and proper time - is an analytical result. By taking a limit of the exact Wightman function we showed analytically that the correlator involves a sum over complex geodesics. We added a sentence to clarify this point.

---

## Round 2 · List of Changes

We made the following changes to address the referee's comments:
Referee 1:
1. We modified the referencing to the paper of Fischetti, Marolf and Wall to make it clear that this was the first paper to point out the role of complex geodesics in correlation functions in de Sitter. We added the sentence ``It was ... higher dimensions.'' near the end of p2.
2. We modified this sentence to ``In addition ... in cosmology.'' to stress that the status of islands in de Sitter space is quite different than for AdS black holes.
3. We added additional explanation around (2.31) to clarify the relation between the integral presented in (2.32), which we used to define the geodesic distance, and the standard expression for the geodesic distance given in (2.31).
4. We improved the explanation the small s-expansion. See the modified paragraph on p10: ``Now $F(x,y;s)$ ... condition (2.27).'' We also added footnote 4 to make it clear that higher powers of s in the heat kernel lead to terms that are suppressed by inverse powers of mass in the propagator.
5. In light of the improved explanation of the small s-expansion we removed this sentence.
6. We added an explanation of this fact in footnote 4.
7. We added footnote 6 to make clear that we are focusing on the Bunch-Davies vacuum and do not consider alpha-vacua.
8. We agree with the referee. In the previous version this statement referred to the wrong equation. We changed the sentence ``It is ... over geodesics'' to refer to (3.1), which does not correspond to a large proper time expansion.
9. We modified Fig. 6 and 7 and distinguished between the Euclidean time coordinate t_E used in Fig. 6 and the Euclidean angle in Fig. 7. The relationship between these two (a priori different) quantities is given around (3.22).
10. We significantly changed the discussion section: ``Assuming the ... firmer footing''. We clarified the difference between the results for the AdS eternal black hole and de Sitter, adding some references to that effect. In addition we clarified our (admittedly) speculative remarks regarding potential non-perturbative corrections that could affect the late-time correlator in de Sitter, including the Euclidean de Sitter black holes instantons. To that effect we also added an additional reference and emphasize that most of the current intuition is based on results in two (or at most three) dimensions.
Referee 2:
1. We agree with the referee that knowing the exact answer was useful to understand which geodesics to include. Nonetheless, the geodesics that need to be included can be determined by a saddle point approximation. We did not show this explicitly in our manuscript, but have added a reference to the paper [23] where this computation was performed explicitly. We added this explanation in the paragraph: ``We would ... is unknown.'' on p16.
2. To add more support to our claim that the geodesic approximation gives a good approximation of the exact result we added Appendix B. Here, we give additional numerical evidence that the difference between the approximation and exact result goes to zero in the limit when the approximation becomes better, as suggested by the referee. In addition, we'd like to stress that our main result - complex geodesics reproducing the exact correlator in the limit of large mass and proper time - is an analytical result. By taking a limit of the exact Wightman function we showed analytically that the correlator involves a sum over complex geodesics. To clarify this point, we added the sentence: ``On top ... Appendix B.'' on p16.

You are currently on this page

---

## Editorial Decision

published